# A macroscopic object passively cooled into its quantum ground state of motion beyond single-mode cooling

D. Cattiaux[1], I. Golokolenov [1], S. Kumar[1], M. Sillanpää[2], L. Mercier de Lépinay[2], R. R. Gazizulin [1], X. Zhou [3], A. D. Armour[4], O. Bourgeois[1], A. Fefferman[1] & E. Collin [1✉]

The nature of the quantum-to-classical crossover remains one of the most challenging open question of Science to date. In this respect, moving objects play a specific role. Pioneering experiments over the last few years have begun exploring quantum behaviour of micron-sized mechanical systems, either by passively cooling single GHz modes, or by adapting laser cooling techniques developed in atomic physics to cool specific low-frequency modes far below the temperature of their surroundings. Here instead we describe a very different approach, passive cooling of a whole micromechanical system down to 500 $\mu$K, reducing the average number of quanta in the fundamental vibrational mode at 15 MHz to just 0.3 (with even lower values expected for higher harmonics); the challenge being to be still able to detect the motion without disturbing the system noticeably. With such an approach higher harmonics and the surrounding environment are also cooled, leading to potentially much longer mechanical coherence times, and enabling experiments questioning mechanical wave-function collapse, potentially from the gravitational background, and quantum thermo-dynamics. Beyond the average behaviour, here we also report on the fluctuations of the fundamental vibrational mode of the device in-equilibrium with the cryostat. These reveal a surprisingly complex interplay with the local environment and allow characteristics of two distinct thermodynamic baths to be probed.

[1] Univ. Grenoble Alpes, Institut Néel - CNRS UPR2940, 25 rue des Martyrs, BP 166, 38042 Grenoble Cedex 9, France. [2] Departement of Applied Physics, Aalto University, FI-00076 Aalto, Finland. [3] IEMN, Univ. Lille - CNRS UMR8520, Av. Henri Poincaré, Villeneuve d'Ascq 59650, France. [4] Centre for the Mathematics and Theoretical Physics of Quantum Non-Equilibrium Systems and School of Physics and Astronomy, University of Nottingham, Nottingham NG7 2RD, United Kingdom. ✉email: eddy.collin@neel.cnrs.fr

Centre-of-mass motion stands out in quantum mechanics[1,2]. It has a central role in quantum models of gravity[3,4] and is at the core of continuous spontaneous localisation (or collapse) models (CSL)[5]. Understanding the quantum behaviour of macroscopic moving objects could thus be the key to the unification of quantum mechanics and general relativity, solving at the same time the long-standing issue of the wave-packet reduction postulate interpretation[6]. In practice, CSL models are most effectively challenged by probing length scales of order 10−100 nm: within the mesoscopic range[5]. Besides, motion is at the core of the basic definition of heat: phonons arise in quantum mechanics as the quasi-particle describing how atoms move. It is therefore important to explore practical ways in which thermodynamics could be probed at the quantum level beyond electromagnetic degrees of freedom[7,8].

Very few setups can detect motion near the Heisenberg limit, that is with the minimum back-action allowed by quantum mechanics[9–11]. In practice, this can be conveniently realised by coupling the mechanical degree of freedom to an optical cavity[12,13]. The tremendous capabilities of optomechanics for force sensing have led recently to gravitational wave detection[14]. One extremely promising technology is the microwave version of optomechanics, where the mechanical element modulates the resonance frequency of an RLC circuit[15,16]. It inherits the properties of conventional optomechanics, is directly compatible with quantum electronics technologies[17,18], with the great advantage of low energy photons being more compatible with cryogenic setups[16,19,20].

Conventionally, a mechanical mode can be said in its quantum ground state when its thermal population $n(T) < 1$, which can be achieved by lowering the temperature $T$. Alternatively, the high degree of control reached within optomechanical systems enables the use of back-action cooling[16,21]. This comes at a cost: the mode is strongly damped by the light field, while the surrounding bath remains warm[22–25]. GHz acoustics has been passively cooled to the ground state using conventional dilution cryostats[26–28]. However, these systems are limited to extremely small centre-of-mass displacements (zero in the case of breathing modes[26,28]), even though they do contain a very large number of individual atoms. Hence, they are not suitable for tests of quantum decoherence and collapse theories[1].

Mechanical modes within micro/nano-mechanical systems with macrocopic masses and which can tolerate large motional amplitudes have resonance frequencies in the MHz range. Their passive cooling, therefore, necessitates sub-milliKelvin temperatures, which is the range attained by ultimate cryogenics: nuclear adiabatic demagnetisation. The ability to measure the system without disturbing it and the demonstration of its thermal equilibrium with the cryostat are both significant challenges[29].

Here, we report on in-equilibrium ground-state cooling of a whole 15-micron diameter aluminium mechanical device resonating at $\omega_{\rm m} = 2\pi \times 15.1$ MHz in its first flexure and coupled to a $\omega_{\rm c} = 2\pi \times 5.7$ GHz microwave cavity, installed on a nuclear demagnetisation cryostat reaching ~500 μK. Remarkably, no signs of thermal saturation are detected in the mechanical properties as shall be discussed below. The mechanical device consists of a 50 nm vacuum-gap capacitor (see Fig. 1) which forms a drum-head[19], and the cryogenic setup consists of a home-made laminar copper nuclear stage mounted on a wet dilution cryostat (see ref. [30] for details). The single-phonon optomechanical coupling $g_0$ for these modes is measured to be ~$2\pi \times 230$ Hz. Single-tone microwave optomechanics is used here only for motion detection[31]. A pump tone (of power $P_{\rm in}$) is applied at an angular frequency $\omega_{\rm c} + \omega_{\rm m}$ (blue-detuned scheme), or conversely at $\omega_{\rm c} - \omega_{\rm m}$ (red-detuned scheme). The mechanical motion generates a sideband peak in the output spectrum at the cavity frequency $\omega_c$. Energy is transferred between the mechanical mode and the microwave field, resulting in

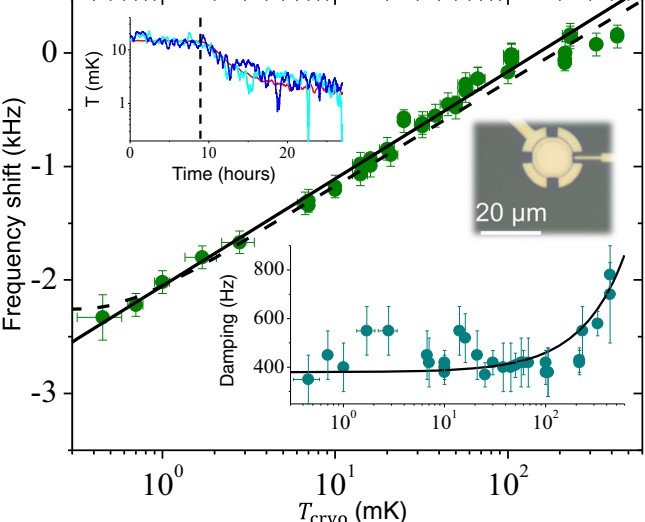

**Fig. 1 Mechanical properties versus temperature.** Main graph: mechanical mode frequency shift $(\omega_{\rm m}[T] - \omega_0)/(2\pi)$ (with respect to $\omega_0/(2\pi) \approx 15.1$ MHz) as a function of cryostat temperature. The black line is a simple Logarithmic fit, whereas the dashed line is the full theoretical fit function (see text). An optical picture of a device from the same batch is also shown. Bottom inset: corresponding mechanical damping $\Gamma_{\rm m}/(2\pi)$. The black line is a linear fit (see text). Top inset: measured temperatures at beginning of cooling in a demagnetisation process (starting at the dashed vertical). The red curve corresponds to the cryostat $T_{\rm cryo}$, whereas the two blue curves are mechanical mode temperatures $T_{\rm mode}$ calculated from the spectrum (from two distinct runs, see text). Error bars correspond to type A uncertainties extracted from the fitting process (Supplementary Note 3).

either enhanced damping of the motion as we increase the drive power (red-detuning), or amplification (blue-detuning). Measurements are performed at the lowest possible powers ($n_{\rm cav} \propto P_{\rm in}$ ~300–600 drive photons confined in the cavity) in order to limit the impact of damping/amplifying. However, driving the system up to the blue-detuned instability (when the damping vanishes), we can show that the mechanical mode exhibits self-sustained motion in the nanometre range[32], which means that these devices are potentially very well adapted for exploring CSL physics. Furthermore, no physical heating of the device can be detected in this range of injected powers, down to 500 μK, which is remarkable. Optical systems are usually limited by their ability to feed-in energy from the photon field, making pulsed experiments mandatory[28,33]. Details on the setup and measurements can be found in Supplementary Notes 1 and 2, respectively.

## Results

**Brute-force cooling.** Formally, three distinct temperatures have to be considered here: the cryostat $T_{\rm cryo}$, the fundamental mechanical mode $T_{\rm mode}$, and the baths $T_{\rm baths}$ directly in contact with it[30]. Demonstrating that thermal equilibrium is maintained across these different subsystems is the key challenge for passive cooling, which we shall address first. Furthermore, characterising the properties of the baths that couple to the mechanical mode is of itself a significant issue. This is our second topic, demonstrating the capabilities of our microwave/microkelvin platform.

In Fig. 1, we present the mechanical frequency shift $\omega_{\rm m}(T) - \omega_0$ (with respect to the high-temperature value $\omega_0$, main graph) and the damping $\Gamma_{\rm m}$ (bottom inset) of the first flexural mode, as a function of cryostat temperature $T_{\rm cryo}$. These parameters are obtained from Lorentzian fits of the measured sideband peak (position and linewidth, respectively). Below 100 mK, the

mechanical damping saturates which is a signature of clamping losses[34,35]. This means that 'phonon tunnelling' dominates energy relaxation, and therefore one of the baths to be considered is the phonons surrounding the drum device. On the other hand, the frequency shift follows a Logarithmic temperature dependence in the whole range, as is commonly observed with low-temperature nanomechanics[36]. This is interpreted as the signature of two-level systems (TLS) defects present in the structure, that couple to the macroscopic mode via the strain field[37,38]. The exact fit expression from theory is shown as the dashed line in Fig. 1 (see Supplementary Note 3 for details)[39]. This leads us to identify a second bath coupled to the mechanics, namely the TLSs. The absence of thermal saturation on $\omega_m(T)$ proves that this bath does cool to the lowest temperatures[30]. As it thermalises through the bulk phonons themselves[39], the surrounding phonon bath has to be cold as well. We can thus infer $T_{TLS} \approx T_{phonons} \approx T_{baths} \simeq T_{cryo}$.

Cooling from ~10 mK to the lowest temperature takes ~2–3 days; the beginning of the process is shown in Fig. 1 top inset. The mechanical mode temperature $T_{mode}$ is inferred from the area of the measured sideband peak using a blue-detuned pumping, as explained below. We clearly see that, on average, the drum-head mode and the cryostat temperatures follow each other, in good equilibrium. Furthermore, we can also see that the mechanics reproducibly displays both very large, and very slow temperature fluctuations. In the following, we will start by discussing the averaged behaviour in detail before going on to describe the properties of the fluctuations.

The sideband signal encodes the position-fluctuations spectrum of the mechanical mode. At low drive powers, the area of the measured peaks with blue and red pumping (Stokes and Anti-Stokes sidebands) are proportional to $P_{in} \times (n + 1)$ and $P_{in} \times (n)$, respectively, with $n(T_{mode}) = 1/(Exp[(\hbar\omega_m)/(k_B T_{mode})] - 1)$ the Bose–Einstein population[20,31]. It is therefore convenient for the experimentalist to define from these areas two 'effective mode temperatures':

$T_{blue} = (n + 1)(\hbar\omega_m)/k_B$ for blue-detuned pumping, measuring Stokes peak,

$T_{red} = (n)(\hbar\omega_m)/k_B$ for red-detuned pumping, measuring anti-Stokes peak.

We will in the following demonstrate thermalisation of $T_{mode}$ by studying $T_{blue}$, $T_{red}$ and their ratio as a function of $T_{cryo}$. In Fig. 2a (right insets), we show sideband spectra measured at different cryostat temperatures and the same drive power (~300 photons), with their Lorentzian fits. For this microwave input, a slight amplifying/de-amplifying exists that is visible from the peak height difference of the high-temperature spectra (blue and red colour for Stokes and anti-Stokes). In order to extract the genuine small-drive limit, we measure the power dependence for each temperature and pumping scheme. This enables us to plot in Fig. 2a (main graph) the $T_{blue}$, $T_{red}$ dependencies with respect to $T_{cryo}$. Lines are theoretical calculations, with no free parameters. At high temperature, $T_{blue} \approx T_{red} \approx T_{cryo}$, as it should; this is essentially the limit of conventional experiments using commercial dilution cryostats[30], and justifies the wording 'effective temperature'.

**Quantum ground state.** However, below typically 10 mK, sideband asymmetry is visible: for blue pumping, $T_{blue}$ saturates while for red $T_{red}$ vanishes (see fits in Fig. 2 the main graph). This effect is a signature of zero-point fluctuations in the microwave cavity[20]. Although $T_{blue}$, $T_{red}$ are no longer simply proportional to $T_{cryo}$, the asymmetry leads to in-built primary thermometry by plotting the ratio of Anti-Stokes over Stokes peak areas[40]. Indeed, in the ideal case of low powers, this reduces to $n/(n + 1)$; for the finite microwave drive amplitude used for the peaks displayed,

the function is renormalized by the ratio of the amplified/de-amplified peak linewidths. This is shown in the top-left inset, Fig. 2a; with the black line being theory with no free parameters. Technical details are given in Supplementary Note 2. The experiment, therefore, demonstrates cooling down to an average population for the first flexure of 0.3 quanta (c.f. the lowest Stokes data point in Fig. 2a). A discussion on the thermal modelling of the device can be found in Supplementary Note 4.

**Mode fluctuations.** Although the long-time average mode temperature always matches that of the cryostat to within measurement uncertainties, we observe strong fluctuations that occur over a surprisingly long timescale, as mentioned earlier. Landauer famously remarked 'the noise is the signal'[41], and in this case, the fluctuations provide important information about the complexity of the thermal environment surrounding the mechanics. To characterise them, we acquire continuously (at fixed drive power, 600 photons, and $T_{cryo}$) sideband spectra using the blue-detuned pumping scheme at a reasonably high repetition rate (typically 1 s per file), and then post-process these data with a sliding average of window 20 mins that produces a fittable peak[30]. From the fit, we can extract peak position, width, and area as a function of time. The area can then be converted (from the calibrated power dependence) into mode temperature (top inset, Fig. 1), or population (Fig. 2b centre)[30]. Width and position shall be discussed later in this article. Although a true reconstruction of the phonon tunnelling statistics is out of the scope of the present paper (the measurement is not at the single-phonon level), what we obtain is a sort of 'smoothed' estimator of the mode occupation number. Furthermore, since the finite measuring power disturbs the system we must always correct for this to obtain the unpumped behaviour. Errors associated with the correction process explain the slight negativity of the lowest histograms in Fig. 2b. These issues are discussed explicitly below and in Supplementary Note 3.

From time traces we compute the fluctuation's spectra (Fig. 2b left). These can be fit by a Lorentzian shape: producing a $1/f^2$ dependence (Brown noise) above a cutoff frequency $1/t_c$ (see full lines). Such a spectrum is reminiscent of an Ornstein-Uhlenbeck process[42]. The question that arises then is: from what thermodynamic bath? We expect fluctuations due to coupling to the phonon bath to have a characteristic timescale $\tau_m = 2/\Gamma_m \approx 1$ ms, the mechanical relaxation time. This timescale is clearly outside of our bandwidth (which is at best about 1 Hz); besides, we observe that the time $t_c \gg \tau_m$ is of the order of 5(±2) hours, and not specifically dependent on temperature (see Fig. 2b left). Therefore, our experiment does not directly probe phonon fluctuations. It is thus natural to consider the second identified bath: TLSs. As shall be discussed below, from the frequency (and damping) fluctuations we infer that TLSs are certainly involved, in a complex and specific way. And indeed, ultra-low-temperature experiments that studied their dynamics report on extremely slow timescales, which are consistent with our $t_c$[43]. A detailed discussion on the statistical treatment is given in Supplementary Note 3.

At low temperatures, the fluctuations can be especially interesting. For example, note the arrow in Fig. 2b centre that points to a time slot where the measured mode population stays at essentially zero over ~5 mins. Knowing that high-frequency modes do cool down to dilution temperatures with no sign of thermal saturation[26,44], higher frequency modes of the structure have to be empty with a high probability. Making an analogy with electronic transport, this result is reminiscent of the one of ref. [45] that demonstrated the absence of any quasi-particles in a superconductor over seconds.

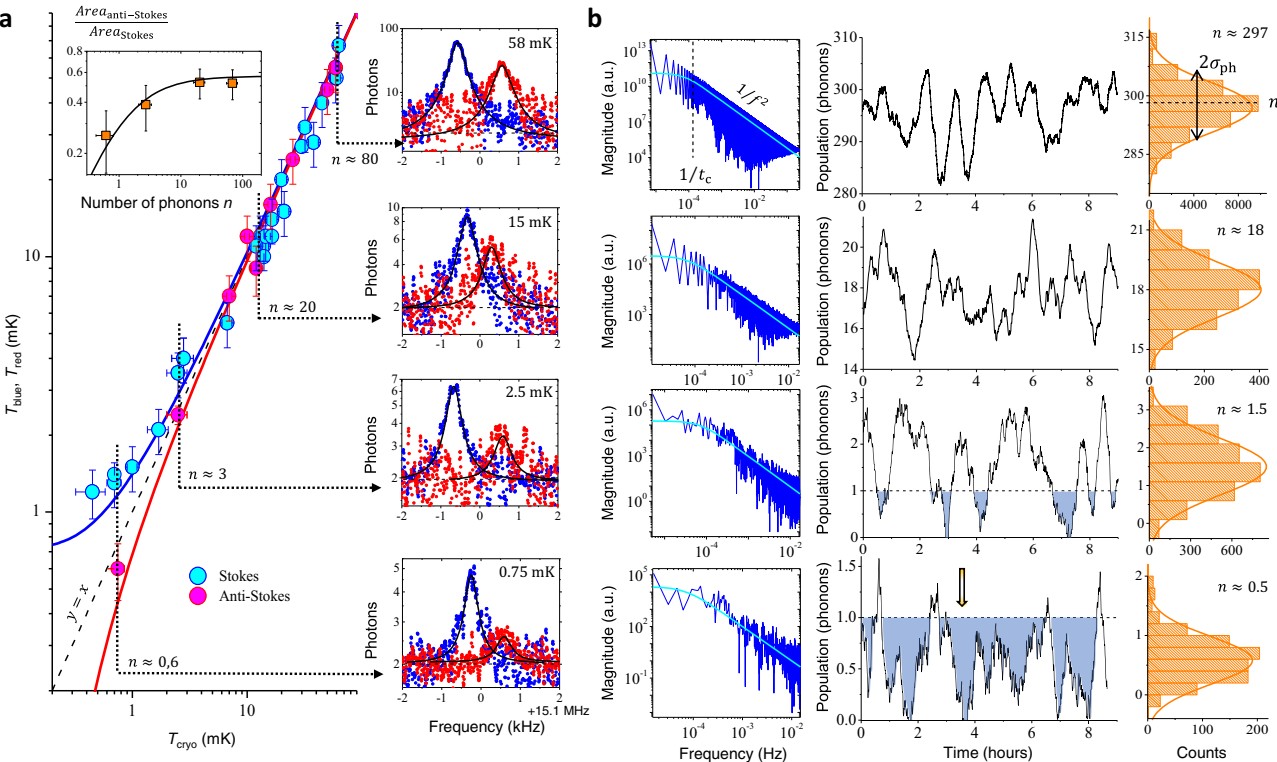

**Fig. 2 Passive ground state cooling. a** Main graph: the peak area of blue and red sideband pumping measurements leads to two different $T_{blue}$, $T_{red}$ estimates, $\propto n+1$ and $\propto n$ (blue and red lines) respectively, a phenomenon known as sideband asymmetry[20]. Right insets: sideband peaks (blue and red colour for Stokes and anti-Stokes) measured for four different temperatures, with their Lorentzian fits (vertically shifted to two-photon baseline for readability). The slight shift in position (with respect to the actual $\omega_m[T]$) is due to a small remnant optical spring due to imperfect tuning of the pump frequency, whereas the difference in peak heights at high temperature is owing to the small amplifying/de-amplifying occurring at the finite input power we use (see text and Supplementary Note 2). The smallest Anti-Stokes peak required 4 days of continuous averaging. Left top inset: ratio of anti-Stokes and Stokes peak areas enabling sideband asymmetry thermometry (the black line is the theoretical expectation). **b** Measurement of in-equilibrium population fluctuations at 220, 13, 1.4, and 0.65 mK (top to bottom) obtained with 20 minutes averaging time per point. Middle: time traces (9 h span) demonstrating very slow (and large, see also inset Fig. 1) fluctuations. The shaded areas delimit the regions below 1 phonon. Left: spectrum (FFT transform of autocorrelation) showing a $1/f^2$ type dependence (full line fit) with a low-frequency cutoff $1/t_c$ (dashed vertical). Right: corresponding histograms from which average $n$ and standard deviation $\sigma_{ph}$ can be defined (with Gaussian fit displayed; the slight negativity for the two lowest graphs is due to the finite precision of the analysis procedure, see text and Supplementary Note 3). Error bars (type A uncertainty) mostly arise from the finite stability of mechanical parameters (frequency, damping, see Fig. 1) and are explained in Supplementary Note 3. Note the arrow on the central lowest time trace that points to a time slot where the measured population drops below our resolution for ~5 mins (see text).

The statistical distributions computed from the time traces look reasonably Gaussian (Fig. 2b right). We can then extract the average population $n$ and its standard deviation $\sigma_{ph}$. This is shown in Fig. 3 (main). Remarkably, we find that the behaviour of $\sigma_{ph}$ is well-described by $0.5\sqrt{n}$, implying a sub-Poissonian mechanism. Note that this fit extends from the truly quantum range up to the classical one, without showing any crossover behaviour. This level of fluctuations is smaller than what we would expect in terms of the fluctuations in the average energy of a system coupled to a simple thermal bath[46,47]. However, this is not surprising since fluctuations associated with the phonon bath are effectively averaged over in the measurement. The observed fluctuations are thus most likely a signature of the (potentially rather complex[48,49]) mode-TLS interaction.

The behaviour of $\sigma_{ph}$ is completely different from frequency and damping noises measured at the same time. They both display true $1/f^2$ spectra (with no low-frequency cutoff observed), and shall thus be characterised for a given time span (so-called Allan deviation)[50]. Their temperature dependence is similar and reversed from the one of phonon population, see inset of Fig. 3 presenting the frequency standard deviation $\sigma_f$: it grows as we cool. This resembles what has been reported on superconducting cavities, and attributed to TLSs

present in the constitutive materials[48]; here, it would be the TLSs-generating frequency shifts from strain-coupling[37,38,51]. Details on the statistical analysis can be consulted in Supplementary Note 3.

Before concluding, we should point out that, as for any measurement, even if the detection power is kept very low it is nonetheless non-zero. There is therefore a finite back-action from the detection scheme onto the mechanical mode: the amplifying/de-amplifying seen in the spectra of Fig. 2a, plus a small cavity noise. It is rather straightforward to correct for these and recalculate average phonon populations[30,31]. It is however perfectly natural to wonder how the finite measurement drive affects the fluctuations themselves, in magnitude, spectral properties and distribution shape. In order to verify experimentally that recalculated fluctuation characteristics do not depend on pump power, we measured a time-trace at half the drive and processed it in the same way: this leads to the orange dot in Fig. 3, which is in agreement with all other data. A thorough discussion on the impact of the measurement scheme can be found in Supplementary Notes 2 and 3.

## Discussion

In conclusion, we have demonstrated ground-state cooling of a whole drum-head device (of mass $5 \times 10^{-14}$ kg), an object that

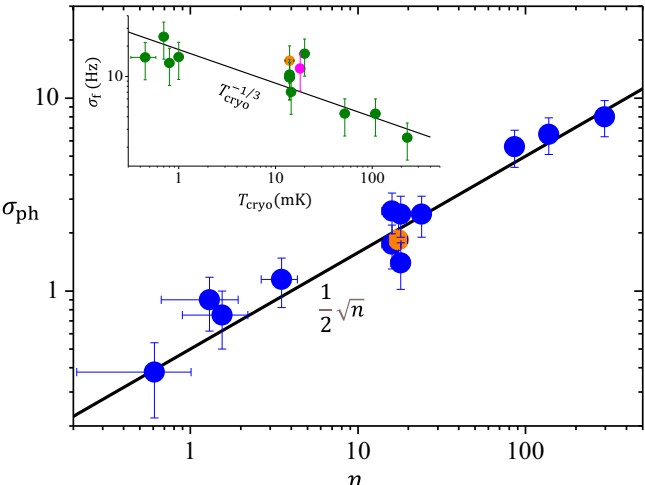

**Fig. 3 Frequency and population standard deviations.** Main: standard deviation of the phonon population extracted from Fig. 2b as a function of the average population. The black line is a $\propto \sqrt{n}$ guide to the eye. Top inset: frequency shift standard deviation calculated over 10 h averaging as a function of cryostat temperature. The line is a power-law fit. The magenta point is inferred from data taken in the self-oscillating regime; in both graphs, the orange dot is obtained with a measuring input power twice smaller (see text). Error bars correspond to type A uncertainties extracted from the fitting process (Supplementary Note 3).

can move its centre-of-mass substantially (typically tens of nm in self-sustained motion of the first flexure), from in-equilibrium properties of the lowest frequency mode. Fluctuations in the occupation of the mechanical mode reveal the complexity of the baths to which it is coupled. The microwave/microkelvin platform developed for this experiment opens the path to a unique regime for experimentalists: quantum thermodynamics can be addressed with phonons through mesoscopic equilibrium properties near the ground state of motion, instead of electrons (or photons)[7,8]. The stochastic nature of quantum heat transport can be studied from one of the collective, macroscopic mechanical degrees of freedom of the device towards the continuum formed by the substrate, through the constriction made by the clamping region. 'Conventional' baths (bulk phonons, TLS) are thermalized; this is mandatory for thermodynamics, but also a unique advantage for the unravelling of new contributions postulated in CSL models[5,52]. This is a feature that is absent for other ground-state cooling platforms focused on macroscopic motion[16,23]. Besides, modal-coupling can be controlled with all modes being cold, down to the lowest frequency one: mechanical decoherence from nearby 'hot modes' (i.e., with a large thermal population) is thus avoided[53,54]. This is a unique advantage of having a whole object ground-state cooled, in comparison with a single working mode as is realised in all other systems[16,23–28].

Passive cooling allows the ground state to be reached while preserving the mechanical Q. This potentially provides much longer mechanical coherence times, enabling unique sensitivities for force sensing to be attained[55]; the figure of merit being $\frac{\Gamma_m}{2\pi} \times n \approx 100$ for us at $500\,\mu\text{K}$, with plenty of scopes for improvements in $\Gamma_m$ (see e.g., refs. [16,56]). The unexpected properties of the fluctuations reported here call for theoretical input on both thermodynamics and the constitutive matter[37,43,51,57]. As for superconducting mesoscopic electronic devices[49], TLS are certainly the key to the understanding of the complex microscopic environment interacting with the mechanical mode.

The present work demonstrates the compatibility between microwave optomechanics and ultra-low temperatures. The next

generation of experiments will incorporate a TWPA (travelling wave parametric amplifier) in order to open the detection bandwidth, potentially down to the phonon relaxation time, while reaching the quantum limit[58]. This would enable the detection of single-phonon jumps in-and-out of the mechanical mode, similarly to electrons in a SET (single electron transistor), which represents the 'holy-grail' of quantum thermal transport experiments. Besides, the microwave circuitry is fully compatible with standard quantum electronics; which means that future developments will also incorporate a quantum bit[18]. This would enable experiments directly focused on the study of quantum mechanical decoherence, as proposed, e.g., in ref. [59]. Such proposals that rely on macroscopic motion can only be implemented on low-frequency devices[60], as opposed to GHz modes.

## Data availability

The data used in this study are available in the openly accessible database following URL: https://cloud.neel.cnrs.fr/index.php/s/CnnYPKn8XHYZgXa.

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

## Acknowledgements

We wish to thank O. Maillet, A. Heidmann, P. Verlot and F. Marquardt for very useful discussions. We acknowledge support from the ERC CoG grant ULT-NEMS no. 647917 (E.C.), StG grant UNIGLASS no. 714692 (A.F.), the STaRS-MOC project from Région Hauts-de-France and ISITE-MOST project (X.Z.). A.D.A. was supported through a Leverhulme Trust Research Project Grant (RPG-2018-213), and M.S. was supported by the Academy of Finland (contracts 308290, 307757, 312057), by the European Research Council (615755-CAVITYQPD), and by the Aalto Centre for Quantum Engineering. The work was performed as part of the Academy of Finland Centre of Excellence programme (project 312057). We acknowledge funding from the European Union's Horizon 2020 research and innovation programme under grant agreement no. 732894 (FETPRO HOT). We acknowledge the use of the Néel Cryogenics facility. The research leading to these results has received funding from the European Union's Horizon 2020 Research and Innovation Programme, under grant agreement no. 824109, the European Micro-kelvin Platform (EMP).

## Author contributions

D.C. ran the experiment and made all measurements. M.S. and L.M. de l'E. designed and fabricated the sample. X.Z. installed and calibrated the whole microwave platform, and R.G. took care of the cryogenics. I.G. and S.K. made preliminary microwave experiments. O.B. analysed and modelled the thermal aspects of the experiment. E.C. designed the experiment. A.D.A., A.F. and E.C. supervised the experiment and the data analysis. All authors participated in the writing of the manuscript.

## Competing interests

The authors declare no competing interests.
