## [Peer Review File · Nature Communications]

Reviewers' Comments:

Reviewer #1:

Remarks to the Author:

I clearly recommend the manuscript to be published in Nature Communications, especially after it has been further improved over the initial version according to the feedback. All my points have been addressed appropriately.

Reviewer #2:

Remarks to the Author:

The bulk of this paper presents a description of a challenging experiment, cooling a MHz-frequency mechanical object to near to its quantum ground state, using only passive cooling. The challenge and the result are quite notable, performing already difficult experiments now in a nuclear demagnetization refrigerator, making these experiments substantially more difficult. The behavior of the system, in the small to many phonon occupation regimes, is studied and reported on, yielding a quite interesting and rich response. These measurements provide some information about the thermalisation process to the different baths that the system interacts with, although many details are not understood.

This work should definitely be published. I am not sure this is the right journal for this work. I feel that the authors are struggling somewhat to identify what makes their results exciting, perhaps to make them more appealing to the broad readership of this scientific journal. As a result, I feel they do a disservice to their results, engaging in hair-splitting exercises that may mislead some readers.

Passive cooling: The abstract could leave some readers thinking this is the first time passive methods have been used to cool a mechanical system to its quantum ground state. This is only corrected in the fourth paragraph, which admits that other experiments have used passive cooling, notably using GHz-frequency resonators, but then includes some statements that are confusing (see below).

Studying a mode versus studying a part of a system: The authors emphasize that they cool the "whole object" of their experiment close to its quantum ground state. It is true that they've cooled and measured the lowest mode of their metal disc to its ground state (and likely cooled all higher modes, although no others are measured), which is indeed remarkable and remarkably difficult. However the distinction seems a bit fine: The metal disc is well connected to a much larger substrate, which presumably has acoustic modes that are quite a bit lower in frequency than the disc (probably a few tens of kHz). The reader is left trying to understand what the fundamental difference is between (a) cooling and measuring a non-fundamental mechanical mode in an system, versus (b) cooling and measuring the fundamental mode in a small part of a system. Note this is aside from the question of passive vs. active cooling, as both have been used in (a), and for the passive approach in (a), of course all higher modes are presumably also in their ground states, although as with (b), only one mode is monitored. What *physical* implications there are for (a) vs (b) is a bit vague; in any case, the mode being studied will exchange energy in some ways with the larger system.

Quantum thermodynamics: Much of the report here deals with some aspects of quantum thermodynamics. Similar results, in a quite different system, were reported in Ref 34, which used pulsed active cooling to cool a mode that is well disconnected from the rest of the system. I am not aware of such studies in other systems, although I am not sure I understand why e.g. GHz-

frequency systems could not be used in the same way, which is implied in the closing paragraphs. The uniqueness of this system, alluded to in the 3rd to last paragraph, is therefore a bit questionable.

Some statements that are new to this version of the manuscript are troubling.

Paragraph 4: "... their bulk mass can be large (on the scale of individual atoms)". Clearly a typo?

Paragraph 4: "... this means they are not suitable for tests of quantum decoherence and collapse theories." I found this statement confusing, perhaps the authors mean to restrict this to tests of quantum decoherence as in Ref 12? Certainly quantum decoherence has been measured in these systems, and quantum measurement theory tested in some ways as well.

Last paragraph: I was confused by the statement regarding the uniqueness of using a qubit to study mechanical decoherence. As mentioned above, this has been done with GHz-frequency resonators. I am not aware of a clear demarcation between what is macroscopic vs. microscopic displacement in that regard, and the volumes involved, at least in SAW resonators, are much larger than here.

Reviewer #1 (Remarks to the Author):

I clearly recommend the manuscript to be published in Nature Communications, especially after it has been further improved over the initial version according to the feedback. All my points have been addressed appropriately.

EC: We thank the referee very much for their careful reading. We indeed believe that the readability of the paper has been improved.

=====

Reviewer #2 (Remarks to the Author):

The bulk of this paper presents a description of a challenging experiment, cooling a MHz-frequency mechanical object to near to its quantum ground state, using only passive cooling. The challenge and the result are quite notable, performing already difficult experiments now in a nuclear demagnetization refrigerator, making these experiments substantially more difficult. The behavior of the system, in the small to many phonon occupation regimes, is studied and reported on, yielding a quite interesting and rich response. These measurements provide some information about the thermalisation process to the different baths that the system interacts with, although many details are not understood.

This work should definitely be published. I am not sure this is the right journal for this work. I feel that the authors are struggling somewhat to identify what makes their results exciting, perhaps to make them more appealing to the broad readership of this scientific journal. As a result, I feel they do a disservice to their results, engaging in hair-splitting exercises that may mislead some readers. EC: Thank you for pointing out that the work “should definitely be published”. We are quite surprised by the (rather new, in respect of what had been exchanged previously) comment of the referee that we “struggle” with our results; in our view, what makes it very “exciting” is evident, as the referee notes: we present “a description of a challenging experiment, cooling a MHz-frequency mechanical object to near to its quantum ground state, using only passive cooling”, something which has not been done before and which opens a range of interesting exciting scientific opportunities which are described in detail at the end of our paper.

Passive cooling: The abstract could leave some readers thinking this is the first time passive methods have been used to cool a mechanical system to its quantum ground state. This is only corrected in the fourth paragraph, which admits that other experiments have used passive cooling, notably using GHz-frequency resonators, but then includes some statements that are confusing (see below).

EC: We believe that the construction of the paper is clear: we claim that we cool an object, not a mode. There is a rational progression in our argumentation to make the point, which deserves some time to introduce the difference between “whole object” and “single mode”. We however understand that, for the non-specialised reader, these words may not carry all the meaning they should. We therefore modified the title of the article stressing out that we go beyond single mode cooling, and clarified also the abstract. Thank you.

Studying a mode versus studying a part of a system: The authors emphasize that they cool the “whole object” of their experiment close to its quantum ground state. It is true that they've cooled and measured the lowest mode of their metal disc to its ground state (and likely cooled all higher

modes, although no others are measured), which is indeed remarkable and remarkably difficult. However the distinction seems a bit fine: The metal disc is well connected to a much larger substrate, which presumably has acoustic modes that are quite a bit lower in frequency than the disc (probably a few tens of kHz). The reader is left trying to understand what the fundamental difference is between (a) cooling and measuring a non-fundamental mechanical mode in a system, versus (b) cooling and measuring the fundamental mode in a small part of a system. Note this is aside from the question of passive vs. active cooling, as both have been used in (a), and for the passive approach in (a), of course all higher modes are presumably also in their ground states, although as with (b), only one mode is monitored. What *physical* implications there are for (a) vs (b) is a bit vague; in any case, the mode being studied will exchange energy in some ways with the larger system.

EC: We are glad that the referee finds our results remarkable. However, we see that some of the language we have used has not been precise enough, leading to confusion. Our claims are that i) the whole of the micromechanical system (drum) and its surroundings are cooled to 500 microK and ii) the fundamental vibrational modes of the drum and its higher harmonics are cooled close to the quantum ground state. To make our meaning as clear as possible we have modified the abstract and now write: "Here instead we describe a very different approach, passive cooling of a whole micromechanical system down to 500 microK, reducing the average number of quanta in the fundamental vibrational mode at 15 MHz to just 0.3 (with even lower values expected for higher harmonics)". Whilst we cannot put all of this in the title, we have modified it to make clear that the focus is on the vibrational modes of our system, the micromechanical drum device: "A macroscopic object passively cooled into its quantum ground state of motion: beyond single-mode cooling". We note that the high-Q of the vibrational modes makes it natural to make a division between them and all the other degrees of freedom which are regarded as a thermal bath characterised by a temperature. The crucial point in our work is that this temperature is just that set by the cryogenics as everything is in thermal equilibrium.

Quantum thermodynamics: Much of the report here deals with some aspects of quantum thermodynamics. Similar results, in a quite different system, were reported in Ref 34, which used pulsed active cooling to cool a mode that is well disconnected from the rest of the system. I am not aware of such studies in other systems, although I am not sure I understand why e.g. GHz-frequency systems could not be used in the same way, which is implied in the closing paragraphs. The uniqueness of this system, alluded to in the 3rd to last paragraph, is therefore a bit questionable.

EC: We again emphasise the basic point that our system, the flexural vibrational modes of micron-sized drum with MHz frequencies, is very different physically from the GHz acoustic resonators studied in [34] and elsewhere. We have added text to the 3rd to last paragraph to make clear the possibility for exciting future work that our specific system offers: "The stochastic nature of quantum heat transport can be studied from one of the collective, macroscopic mechanical degrees of freedom of the device towards the continuum formed by the substrate, through the constriction made by the clamping region". We also would like to refer the referee to the detailed response we gave in the previous round to Referee #3 who asked about the connection with Ref [34] in their comments. The limitations of Ref. [34], and differences with what is obtainable in our system are described at length.

Some statements that are new to this version of the manuscript are troubling.

Paragraph 4: "... their bulk mass can be large (on the scale of individual atoms)". Clearly a typo?

EC: what is meant is the difference between the **mass of all of the atoms** of the device as opposed to

the “effective” mass of the mode, which can be reasonably small. We rephrased it in the new version “even though they do contain a very large number of individual atoms”. Thank you.

Paragraph 4: "... this means they are not suitable for tests of quantum decoherence and collapse theories." I found this statement confusing, perhaps the authors mean to restrict this to tests of quantum decoherence as in Ref 12? Certainly quantum decoherence has been measured in these systems, and quantum measurement theory tested in some ways as well.

EC: We have modified the relevant sentence in the text to make our specific meaning, relating to the decoherence of mechanical motion that involves a large centre-of-mass motion (see text quoted in the previous point). Whilst many aspects of quantum theory have been explored in micromechanical systems, the way in which the decoherence of superpositions of states involving spatial separation of a (centre-of-) mass that is macroscopic on the scale of atoms has not been explored. The point being that there are questions about how decoherence and other mechanisms such as wavefunction collapse occur for these specific types of quantum states, and the ultra-cold in-equilibrium system we study would offer specific advantages in doing so. Although we pick up on this issue at the end of the paper and seek to reinforce the message that the relevance is for exploring decoherence related to macroscopic motion, space constraints do not allow us to explain all of the issues, which we agree can be subtle, in full detail. However, we believe that the references we cite do provide the necessary background.

Last paragraph: I was confused by the statement regarding the uniqueness of using a qubit to study mechanical decoherence. As mentioned above, this has been done with GHz-frequency resonators. I am not aware of a clear demarcation between what is macroscopic vs. microscopic displacement in that regard, and the volumes involved, at least in SAW resonators, are much larger than here.

EC: The uniqueness is in the protocol proposed in Ref. [62], which has nothing to do with e.g. Ref. [32] and involves using a qubit to produce (and probe the coherence of) superpositions of spatially separated states of a macroscopic (on the scale of atoms) mass. The proposal is built around the dispersive coupling to a low frequency mechanical device with a centre-of-mass that can be displaced significantly. Precisely the kind of micromechanical system we have here. What the referee refers to are experiments where GHz modes can exchange energy directly with a qubit and consequently involve very different classes of quantum states. Again, these GHz modes (even the SAW ones, as explained in our previous responses) sustain extremely small motion as opposed to large flexural objects (like drums). In this respect, probing CSL at the 10 – 100 nm range is clearly out of question for GHz modes. Again, we believe that space considerations mean that we cannot enlarge on this in the main paper, but we believe that the interested reader will find all the relevant detail within the references provided.

=====

Reviewer #3:

No comments to the Authors.

EC: Thank you.